# Microbial Metabolites and Cardiovascular Dysfunction: A New Era of Diagnostics and Therapy

**DOI:** 10.3390/cells14161237

**Published:** 2025-08-11

**Authors:** Jitendra Kumar

**Affiliations:** Department of Biochemistry and Molecular Biology, Carver College of Medicine, University of Iowa, Iowa City, IA 52242, USA; jitendra-kumar@uiowa.edu or jitendramdu@gmail.com

**Keywords:** cardiovascular diseases, atherosclerosis, gut microbiota, microbial metabolites, endothelial dysfunction, therapeutic targets

## Abstract

Cardiovascular diseases (CVDs) pose a significant threat to human life and mortality worldwide, encompassing a variety of conditions that affect the heart and blood vessels. These diseases are influenced by both genetic and environmental factors, which play a critical role in their development. Recent research has highlighted the importance of gut microbes—the diverse community of bacteria in the gastrointestinal tract—that function as a “super organ” within the human body. These microbes have a remarkable impact on metabolic pathways and are increasingly recognized for their role in serious conditions like CVDs. They contribute to metabolic regulation, provide essential nutrients and vitamins, and help protect against diseases. Various internal and external factors influence the dynamic relationship between the human host and gut microbiota, thereby regulating overall metabolism. This review explores the complex connection between gut microbiota and microbial metabolites—such as short-chain fatty acids (SCFAs), bile acids (BAs), and trimethylamine N-oxide (TMAO)—and their potential influence on the development and progression of CVDs. We also examine the interaction between dietary interventions and gut microbes in the context of conditions including atherosclerosis, obesity, type 2 diabetes, heart failure, hypertension, atrial fibrillation, and myocardial infarction. Gaining a deeper understanding of the gut microbiota’s role in maintaining physiological balance creates exciting possibilities for identifying novel diagnostic biomarkers and therapeutic targets for treating CVDs. This knowledge offers hope for early disease prediction, improved clinical management, and innovative treatments.

## 1. Introduction

Globally, CVDs are major contributors to morbidity, disability, and mortality, with rising incidences in the recent years. Therefore, they pose a major concern for all developing countries regarding health and socioeconomic challenges. Gut microbes and their metabolites might have therapeutic potential for CVDs and that needs to be connected. The common risk factors for the development of cardiovascular diseases include obesity, hypertension, and type 2 diabetes (T2D) [1]. All three factors are lifestyle and environmental, and might be prevented by monitoring diets, maintaining gut microbes, and engaging in physical activities. Humans have a plethora of microbiotas, which are present throughout the body, including the skin, mouth, stomach, and intestines, in a mutualistic relationship [2].

There is a wide diversity of bacteria in our foregut and midgut, which is regulated by our dietary habits. An equilibrium of different types of bacteria generally benefits human health, while disequilibrium may lead to more severe health problems [3,4,5,6,7,8,9,10,11]. The commonly found species of bacteria in the human gut include Firmicutes, *Actinobacteria*, *Bacteroidetes*, *Verrucomicrobia*, and *Proteobacteria* [12,13,14,15,16].

The digestion of food is regulated by the microbial community. It helps in converting complex compounds into simpler ones through various metabolites and helpful enzymes [17]. They benefit humans by aiding in synthesizing vital components such as vitamins B and K, fermenting different dietary products like short-chain fatty acids (SCFAs) and secondary bile acids (BAs), and aiding in the metabolism of bile, xenobiotics, and sterol compounds [18,19,20]. The gut microbiome is central in modulating gut function and synthesizing, controlling, and suppressing essential metabolic enzymes closely associated with various health and disease mechanisms [21,22,23]. Hence, it would be right to consider gut microbiota as body’s “super organ” because it is closely connected to our overall health. With aging, the balance of intestinal microbes and human metabolism are affected. Multiple studies utilizing metagenomics, metabolomics, and metalloenzymes have contributed to our understanding of the gene sequences of gut microbial communities [24,25,26,27]. It shows that human health depends on gut microbes, and that microbes involved in the pathophysiology and metabolism pathways where here it plays a role in human health development through nutrients process. Numerous studies have exhibited the vital role of gut bacteria in regulating different body parts, leading to terms such as the gut–liver axis, gut–brain axis, and gut–kidney axis being coined by other research groups [28,29,30,31,32]. In addition to the positive role played in the maintenance and regulation of various metabolic processes, the gut microbes are intricately involved in the development of many diseases, such as colorectal cancer, cerebral ischemia–reperfusion injury, diabetes mellitus type 1 and 2, metabolic syndrome, and CVDs [33,34,35,36,37,38,39,40,41,42,43,44,45,46,47,48,49,50,51,52,53,54,55,56,57,58]. They have the potential to impact the function of various metabolic pathways in the body’s organs; thus, their influence is extended beyond the development of the disease (Figure 1). Gut dysbiosis plays a significant role in development of CVDs such as heart failure and myocardial infarction [59,60,61,62,63,64,65,66,67,68]. For example, *Desulfovibrio desulfuricans* is involved in TMA/trimethylamine production, while Firmicutes and Proteobacteria are linked to choline consumption [63,69]. Metabolites like lipopolysaccharide (LPS), TMAO, SCFAs, secondary BAs, and uremic toxins such as *p*-cresol sulfate and indoxyl sulfate may also play a role in cardiovascular function [63,69,70,71,72,73,74,75,76,77]. This review emphasizes our present comprehension, presenting noteworthy proof of the intimate connection between human intestinal microorganisms and cardiovascular conditions. It also aims to assess potential therapeutic approaches that can influence gut flora’s diversity, function, and metabolites.

## 2. Key Gut Microbial Metabolites and CVDs

The gut microbiota’s metabolites have been linked to various human metabolic processes. Some of these metabolites have been found to influence the development of different types of cardiovascular diseases, as outlined in Table 1.

### 2.1. Short-Chain Fatty Acids

The fermentation of dietary fibers, primarily polysaccharides, in the human body by gut microbes is essential in digesting complex carbohydrates since the human body’s physiological conditions do not facilitate this process [78]. The composition of gut microbiota significantly influences fiber fermentation, resulting in glycolysis that breaks down glucose into pyruvate, which produces short-chain fatty acids (SCFAs) [79,80]. The most common SCFAs in the human body are acetate, propionate, butyrate, and isovalerate [81,82]. Acetate is vital in gluconeogenesis and reduces appetite through a central homeostatic mechanism. Additionally, acetate is essential in synthesizing low-density lipoprotein (LDL) cholesterol. Propionate has been reported to affect intestinal gluconeogenesis, while butyrate is an energy source for intestinal epithelial cells, contributing to colonic homeostasis care [83,84]. Furthermore, SCFAs are potentially involved in immunomodulatory action, regulating anti-inflammatory responses, producing mucus in the gastrointestinal tract, preventing intestinal inflammation, and maintaining intestinal barrier integrity [79,84].

SCFAs have the potential to modulate microbiota that in turn regulates the cardiovascular diseases (CVDs). Gut microbiota, which produce sodium butyrate and propionate, relate to the function of the prorenin receptor-mediated intrarenal renin–angiotensin system. According to a study, a high-fiber diet and acetate supplements help balance blood pressure and cardiac fibrosis by improving gut dysbiosis and helps in the maintenance of *Bacteroides acidifaciens*. Bacteria that produce propionate and butyrate (such as *Roseburia intestinalis*) protect from the risk of hypertensive cardiovascular damage and aortic atherosclerotic lesions [85,86,87,88]. SCFAs are known to interact with immune and endocrine cell receptors in the nervous system, kidneys, and blood vessels; for example, SCFAs regulate blood pressure with the help of Olfr78 (olfactory receptor 78) and GPR41 (G-protein-coupled receptor 41) in small resistance vessels [88,89], and both receptors are essential for maintaining blood pressure regulation under healthy conditions. GPR41 acts as a hypotensive protein to dilate resistance vessels in an endothelium-dependent manner, while Olfr78 functions as a hypersensitive protein [89,90]. The direct activation of GPR1 has been reported to impact inflammation and enteroendocrine regulation.

It has been reported that microbiota-based SCFAs are immunomodulatory in attenuating oxidative stress and maintaining the functional system [91]. Butyrate produced by gut microbiota has been shown to slow atherosclerosis progression due to its anti-inflammatory function. Butyrate also regulates luminal gut oxygen levels by activating peroxisome proliferator-activated receptor gamma in colonocytes. There is a role of acetate and butyrate in improving rat aortic endothelial dysfunction by increasing the bioavailability of nitric oxide, which occurs due to the activation of GPR41/43 for butyrate only [90]. Other GPCRs involved are GPR43 and GPR109A, or histone deacetylases that act via Tregs (immune-suppressive regulatory T cells), interleukin-18 secretion in the intestinal epithelium, and interleukin 10-producing T cells. GPR109A helps in the prevention of colon cancer initiation. SCFAs and microbiota have beneficial effects in preventing cardiac hypertrophy, fibrosis, vascular dysfunction, obesity, and hypertension [86]. Based on a study involving TLR5-knockout mice, it was reported that short-chain fatty acids (SCFAs) have been linked to metabolic syndrome [92]. The dysbiosis of gut microbiota leads to excessive and prolonged SCFA production, which triggers metabolic syndrome [93,94]. TLR5 is a receptor for flagellin, which plays a role in maintaining microbiota balance. This research indicates that the presence of *Phascolarcto bacterium*, *Veillonellaceae*, and *Proteus mirabilis* is strongly associated with obesity, and their alteration leads to the increased production of acetate and propionate [95]. It might be concluded that the different gut microbiota have an essential role in the production of SCFAs that regulate metabolic syndrome, in turn, affecting CVDs.

### 2.2. Bile Acids

In humans, BAs are hydroxylated and saturated steroids. Bile acids have multiple functions, such as microbial activity, acting as signaling molecules, and stimulating metabolism. Primary bile acids (found in the liver), such as cholic acid (CA) and chenodeoxycholic acid (CDCA), are synthesized from cholesterol in a multistep process in the liver. Several catalytic enzymes, including cholesterol 7a-hydroxylase (CYP7A1), sterol-27-hydroxylase (CYP27A1), and oxysterol 7a-hydroxylase (CYP7B1), are involved in this process and play a crucial role in the emulsification, digestion, and absorption of fat-soluble molecules, dietary fats, and vitamins [96]. The expression of catalytic enzymes bile salt hydrolases (BSHs) is controlled by the human intestinal commensal microbes, e.g., the Gram-positive *Bifidobacterium*, *Clostridium*, *Enterococcus*, and *Lactobacillus* and the Gram-negative *Bacteroides*. These gut microbiota in the intestine convert the primary BAs into secondary BAs with the help of bacterial salt hydrolase activity (which removes -OH) [97,98]. The 7-dehydoxylase enzyme, sourced from *Clostridium* and *Eubacterium*, is vital for converting primary BAs into secondary BAs. In the small intestine, conjugated BAs are deconjugated via microbial hydrolases, reducing resorption and improving cholesterol exclusion. Microbially metabolized bile acids in the blood regulate host metabolism, which can lead to cardiovascular diseases. BA modification by intestinal microbes occurs before returning to the liver (re-conjugation) and to the circulation. The FXR (Farnesoid X-activated receptor) bile acid receptor is a critical sensor [99]. FXR activation inhibits atherosclerosis injuries in atherosclerosis-prone mice. Deletion of FXR in ApoE-/- mice initiates plaque buildup and increases the severity of lipid metabolism [99,100,101]. However, double-mutation (FXR/ApoE or FXR/LDL) mice showed a reduction in plaque buildup and plasma low-density lipoprotein [102]. The TGR5 (Takeda G-protein-coupled receptor), a BA receptor, activated by secondary BAs, causes decreased intraplaque inflammation, reduced plaque macrophage content, and decreased lipid loading to diminish vascular lesion buildup. Many studies have reported on the mechanism of bile acids in cardiovascular disease pathogenesis [103].

The risk of atherosclerosis or plaque formation increases if the cholesterol circulation level decreases due to decreased primary and secondary bile acid levels. In chronic heart failure patients, the level of primary BAs was reduced while the level of secondary BAs increased. Microbiome-based bile acids can potentially cause atherosclerosis and myocardial infarction (MI) through different types of bile acid receptors by reducing bile acid synthesis and increasing plasma LDL cholesterol [104,105].

### 2.3. Trimethylamine (TMA)

The human gut microbes initiate TMAO co-metabolite production via TMA hepatic oxidation with the help of flavin monooxygenase enzymes (FMOs, mainly FMO3) [106]. Initially, TMAO was generated within the bacterial metabolism of dietary choline [107], L-carnitine [108], and phosphatidylcholine [106]. It is an organic compound metabolized by intestinal microbes into TMA in the blood. CVD risk is higher when the hematic concentrations of TMAO are increased [108]. In vivo, its high concentration relates to the impairment of calcium signaling pathways, finally stimulating prothrombotic phenotype and platelet hyper-reactivity [109]. It was reported that, in mice, a microbial composition alteration due to chronic dietary L-carnitine increases TMA and TMAO production and ultimately initiates atherosclerosis plaque enhancement. However, when the supplement diet reduces and alters intestinal microbiota composition, it inhibits TMAO production and decreases atherosclerosis lesions [108,110,111]. In the human gut, seven types of microbiotas, such as *Anaerococcus hydrogenates*, *Clostridium hathewayi*, *Clostridium sporogenes*, *Clostridium asparagiforme*, *Escherichia fergusonii*, *Providencia rettgeri*, and *Proteus penneri*, are observed to regulate the TMA lyase *Cut C/D* [107,112,113]. It consists of functional microbial genes that are accountable for choline-related TMA transformation. Primarily, gut bacteria regulate TMAO production through different mechanisms like foam cell formation (HSP70 or HSP60, SR-A1, ox-LDL), inflammation (IL-6, COX-2 (cyclooxygenase 2) [108,111,114], intracellular adhesion molecule 1, MAPK, NF-iB), lipid metabolism (RCT (reverse cholesterol transport)), and platelet hyper-reactivity and thrombosis (ADP (adenosine diphosphate), collagen, and thrombin) [115]. Based on different in vivo and in vitro experiments, it has been reported that TMAO has a significant potential to affect the inflammation of aortic endothelial cells, oxidative stress, and inflammatory response, which increases the CVD risk. Choline or TMAO in animal studies shows the connection between gut microbiota and TMAO, which causes CVDs. Studies also indicate a rational relation between plasma TMAO and CVD possibility. For the diagnosis of and the prediction of the possibility of different CVDs like MI (myocardial infarction), stroke, HF (heart failure), and peripheral artery disease, the high value of circulating TMAO should be used as a biomarker. A reduced use of carnitine, choline, and lecithin-rich diets prevents the onset of CVDs. Some natural compounds like DMB (3,3-dimethyl-1-butanol), present widely in red wines, vinegar, and some grape seed oils, prevent the TMA lyase activity of microbial choline [116,117]. Its effects are also observed in the atherosclerotic lesions of Apoe-/- mice [106]. Other TMA lyase inhibitors (like choline) that reduce the plasma TMAO levels are bromomethylcholine, chloromethylcholine, FMC (fluromethylcholine), and IMC (iodomethylcholine). This means, significance of healthy gut microbiota in regulating TMA/TMAO is important for the regulation of good health.

### 2.4. Other Metabolites

Microbial metabolites derived from aromatic amino acids (AAAs) impact cardiovascular health. These aromatic amino acids are added to diets primarily through dietary proteins found in chicken, fish, pork, beef, and goat. It has been reported that the metabolites, such as phenylacetylglutamine (PAG) and others, have been closely linked to CVDs through various biochemical pathways, thus highlighting the critical role played by gut microbiota in human health as well as disease [118,119,120,121]. Notably, research has indicated that microorganisms in the human gut, specifically *Clostridium sporogenes*, are responsible for producing metabolites from AAAs [118].

The studies have identified microbial metabolites such as PAG and their potential association with CVDs via the GPCR signaling pathways [119]. Similarly, reduced levels of microbe-derived tryptophan metabolites in the blood plasma of individuals with atherosclerosis have been documented, indicating a possible link to cardiovascular health outcomes [122]. In the rodent system, it has been reported that there is an involvement of tyrosine and phenylalanine microbial metabolites in myocardial infarction. The other intestinal microbe-related metabolites such as indoxyl sulfate (IS) and *p*-cresol sulfate (PCS) have shown therapeutic potential in diagnosing the onset of CVDs. However, there are conflicting findings regarding their direct connection to cardiovascular health [123,124,125,126].

In the realm of human health, the impact of plant-derived polyphenols on overall well-being has gained significant attention. These organic molecules, found in foods like tea, coffee, apples, onions, cocoa, and citrus fruits, are recognized for their potential in preventing T2D and CVDs. The absorption, circulation, and transportation of polyphenols within the body are influenced by gut microbiota, underscoring the intricate interplay between diet, gut microbiota, and human health. Notable subclasses of dietary polyphenols, including anthocyanins, flavanols, flavanones, and hydroxycinnamates, have been the focus of numerous research due to their potential health benefits and impact on disease prevention [127,128].

## 3. Gut Microbial Dysbiosis and CVDs

The human gut, a bustling hub of non-pathogenic microorganisms, is a significant regulator of a wide range of CVDs. These encompass hypertension, T2D, obesity, atherosclerosis, and heart failure (Figure 2). A multitude of studies have explored the complex relationship between gut microbial dysbiosis and these specific cardiovascular conditions, unveiling potential mechanisms and implications for treatment and prevention strategies.

### 3.1. Hypertension

Dysregulated blood pressure is a global disease and a significant cause of cardiovascular disease. The changes induced due to hypertension initiate changes in the intestinal environment that induce the dysbiosis of microbial communities residing in the gut. The dysbiosis of gut bacteria and related metabolites, due to the increased *Firmicutes/Bacteroidetes* ratio, the abundance of *Klebsiella* spp., *Parabacteroides merdae,* and *Streptococcus* spp., the diminished population of acetate/butyrate metabolite bacteria, SCFAs, and TMAO are well connected with the seriousness of hypertension [129,130]. A disruption of the intestinal barrier along with the elevation of harmful gut microbiota result in chronic inflammation in hypertensive patients. These harmful bacterial genera increase the pro-inflammatory signals and reduce the immunity of the intestine in hypertensive patients.

Mell et al. (2015) [129]. present a significant difference in microbiota in salt-sensitive and salt-resistant strains in Dahl rats [129]. It was observed that the angiotensin-II-infused germ-free mice, angiotensin II-induced vascular dysfunction, and hypertension are regulated by mice gut bacteria [131]. An increased level of *Odoribacter* (butyrate-producing genus) relates to heavy-weight hypotensive women [132]. Using the metagenomics data based upon the different fecal samples of healthy, prehypertensive, and hypertensive individuals suggested the role of gut microbiota in hypertension pathogenesis. The elevation in the *Klebsiella* and *Prevotella* bacterial population in prehypertensive and hypertensive individuals was reported [133]. Numerous studies in mice, rats, and germ–free rats have reported that the unbalanced composition of gut bacteria and inflammation due to a dysfunctional sympathetic nervous system–gut interaction [134,135,136]. Hypertension pathogenesis is changed with the alteration in the relevant gut flora, which shows that intestinal microbiota may have a substantial role in hypertension treatment by using potential probiotics, FMT (fecal transplant), and antibiotics.

### 3.2. Heart Failure (HF)

HF is a disease in which heart blood pumping efficiency is reduced and does not supply sufficient blood and oxygen as per the body’s needs. Many factors contribute to HF, and gut microbiota is one of them. It involves regulating the immune and inflammatory responses that favor the HF pathophysiological mechanisms [137]. Gut dysbiosis involves reduced cardiac efficiency and raised systemic congestion that leads to ischemia and edema of the intestine mucosa, improves bacterial translocation, and increases circulating lipopolysaccharide levels (endotoxins), which play a role in the underlying inflammation in HF patients [138,139,140]. Moreover, *Clostridium difficile* infection is reported in the HF patients. Furthermore, the sequencing data of microbial diversity of fecal samples of HF patients showed a diminution of many butyrate-producing bacteria and *Dorea longicatena* and *Eubacterium rectale*. However, pathogenic *Campylobacter*, *Shigella*, and *Salmonella* bacteria are abundant [141]. In addition, based on metagenomics and metabolomics data, it is shown that some HF patients reported a significant decrease in *Blautia*, *Collinsella*, and *Faecalibacterium prausnitzii* [137,142]. Paisini et al. (2016) [141] study shows that chronic heart failure patients have higher intestinal permeability and a higher number of gut bacteria (in feces) than control individuals [141]. Different animal studies have demonstrated the richness of different types of fecal flora in HF animals. This indicates that HF may cause the dysbiosis of gut microflora that supports the progression of the disease [143]. TMAO is a microbiota-associated metabolite, and few studies have demonstrated that increased levels of circulating TMAO are predictors of acute and chronic heart patients compared to healthy controls [144]. This metabolite is also associated with other CVDs like left ventricular dilatation, myocardial fibrosis, ventricular remodeling, and cardiac hypertrophy [145].

### 3.3. Acute Myocardial Infraction (AMI)

Arterial blockage is a major health issue. Based on this, coronary artery disease is divided into stable coronary artery disease (CAD) and acute coronary syndromes (ACS) [146]. The leading cause of CAD is a two-sided, blood supply–demand event mismatch association with myocardial ischemia. In contrast, ACS clinically shows AMI, unstable angina pectoris (UAP), or sudden cardiac death (SCD) [147]. After outstanding achievements in medical fields, the early prediction of AMI is still a big challenge. Many diagnostic biomarkers are available to diagnose AMI, but reliable and effective techniques are still unavailable. Therefore, more research is needed for an accurate biomarker to diagnose clinical AMI. The primary biomarkers for AMI are cTnI (cardiac troponins I) and cTnT (cardiac troponins T). The main limitation of cTnI is that it is usually high in both sCDA and MI patients [148,149]. Different studies have demonstrated that, in AMI patients, exosomal miR-1, mir-4507, mir-1915-3p, and miR-3656, CK (creatine kinase) and CK-MB (creatinine kinase myocardial band) are drastically higher when compared with sCAD and healthy persons. However, their forecast precision is not satisfactory [150,151,152]. Therefore, more biomarkers with high sensitivity and specificity for AMI are needed. Numerous animal studies demonstrated intestinal microflora’s role and related metabolites in AMI occurrence. Different bacterial families are significantly higher in AMI rats than the SHAM controls, such as *Dethiosulfo vibrionaceae*, *Eubacteriaceae*, *Lachnospiraceae*, and *Syntrophomonadaceae* and the genera *Tissierella* and *Soehngenia*. In comparison with healthy individuals, patients with AMI harbor plenty of *Acidaminococcus*, *Butyricimonas*, *Desulfovibrio*, and *Megasphaera* [153,154,155,156,157,158,159,160,161,162,163,164]. The study of Liu et al. (2019) [155]. showed that the typical changes in bacterial co-abundance group (CAG) in coronary artery disease (CAD) were dominated by *Roseburia*, *Klebsiella*, *Clostridium IV, and Ruminococcaceae* [155]. In contrast, a recent study reported that *Alistipes*, *Streptococcus*, *Ruminococcus*, and *Lactobacillus* are efficient in differentiating AMI from sCAD, and their prognostic competence has been confirmed in an independent cohort of CAD patients [156]. Meanwhile, a high level of TMAO in blood plasma and proatherogenic and prothrombotic bacterial metabolites relate to different adverse symptoms in ACS patients. Furthermore, the tryptophan metabolites of the kynurenin pathway have been linked to a higher likelihood of AMI in patients with suspected sCAD [157,158]. Based on these studies, intestinal microflora strongly correlates with CAD patients, altering the blood metabolite content and composition. Hence, intestinal microflora and serum metabolites have been identified as potential diagnostic markers of AMI.

### 3.4. Atherosclerosis

It is a chronic inflammatory disease leading to the initial endothelial injury or dysfunction of vascular cells and the development of plaque in the arteries due to the buildup of LDL bits. Different types of cells, like endothelial cells, macrophages, foam cells, and leukocytes, accumulated lipids like cholesterol and triglycerides which involved in a plaque formation. Atherosclerosis risk is generally high in individuals with hypertension, obesity, diabetes mellitus, and hypercholesterolemia [159,160]. Ott et al. (2006) [161] reported *Staphylococcus* species, *Proteus vulgaris*, *Klebsiella pneumoniae,* and *Streptococcus* species in atherosclerosis plaques and the gut of the same individual. These results recognized the possibility of the presence of gut microbiota in the process impacting plaque formation and stability and causing atherosclerosis [161]. Atherosclerosis is a common chronic inflammatory CVD that affects the coronary and peripheral arteries. Gut microbiota and its metabolites play a crucial role in the development of CVDs. Increased Firmicutes/Bacteroidota (F/B) ratio in gut is an indicator of CVDs.

Furthermore, the presence of oral microbiota is also reported in the atherosclerotic plaque in humans [162]. Karlsson et al. (2012) [163] used metagenomic sequencing to study the composition of gut bacteria in stool samples and reported the dysbiosis of gut microflora in patients with stable plaques versus unstable plaques. They found a diminished level of butyrate-producing bacterial genus *Roseburiam* in the unstable plaque patients, and symptomatic atherosclerotic patients harbor a greater abundance of *Collinsella* compared with healthy controls [163]. Gut metagenomes, coding genes for pro-inflammatory peptidoglycan synthesis are abundant, and the phytoene dehydrogenase level is low in patient. The gut microbiome of the patients shows a low production of anti-inflammatory serum levels of β-carotene [43]. Other studies on humans show that the gut microbiome is possibly involved in the pathogenesis of atherosclerosis through its role in lipid metabolism. Middle-aged men with improper total cholesterol and LDL-C values harbor more *Prevotella* and lower levels of *Clostridium* and *Faecalibacterium* [164]. However, other metagenome-wide association studies reported a lesser number of Bacteroides and *Prevotella* and the enrichment of Enterobacteriaceae (including *Escherichia coli*, *Enterobacter aerogenes*, and *Klebsiella* spp.) and *Streptococcus* spp. in atherosclerotic cardiovascular disease patients [165]. There is a relationship between various gut microbiome metabolite levels, such as TMAO, BA, and SCFA, and the pathogenesis of CVD. Different factors and CVD symptoms are involved in microbial dysbiosis, such as the mutation of TMAO (toxic metabolites) and pathogenic bacteria [165,166,167,168]. To understand the specific gut microbiome biomarkers of atherosclerosis development and their mechanism, more investigation is needed.

### 3.5. Atrial Fibrillation (AF)

AF is not just a local issue but a global one, affecting people worldwide. It is one of the most ubiquitous and extensive cardiac arrhythmias in clinics. The fast and uneven atrial beating represents an abnormal heart rhythm [169]. Ischemic stroke (cerebral infarction) and cardiac insufficiency risk factors are increased in the AF. Based on clinical and pathological systems of AF, AF decreases the end-diastolic volume and causes fast atrioventricular desynchrony, a fast ventricular rate, and a loss of atrial systolic pump function, leading to a decline in cardiac systolic function and heart failure. Furthermore, in ischemic stroke patients, AF enhances the mortality, recurrence rate, and infirmity rate [170,171]. It is a global disease, and due to its infirmity and morbidity rate, AF generates a serious issue for society and affects the excellence of human life. Therefore, the prevention of AF is needed. The human gut microflora plays a vital role in the onset of AF. Zuo et al.’s (2019) [172] study showed that the composition of AF patients’ gut flora significantly differs from that of normal individuals. In AF patients, there is an abundance of *Enterococcus*, *Ruminococcus*, and *Streptococcus*. However, decreased levels of *Alistipes*, *Oscillibacter*, *Bilophila*, and *Faecalibacteriumare* were reported [172]. Moreover, another study based on metagenomics and metabolomics reported the short and long psAF (persistent AF) and highlighted the relationship of gut bacteria with psAF maintenance. They found significantly different types of gut bacteria in both short- and long-term psAF patients, with an elevated microbial dysbiosis, diverse composition, and inconsistent structure. They also observed modifications in related bacterial metabolites and disorders in gut microbial function [173]. LPS (lipopolysaccharide) produced from bacteria reacts as an endotoxin in human blood by stimulating pro-inflammatory cytokines that reduce the L-type Ca^2+^ channel expression and shorten ERP (effective refractory period). ERP shortening is also observed due to the other bacterial metabolites like choline that activate the acetylcholine-dependent potassium channels at high concentrations. LPS is also connected with different CVDs like atherosclerosis, heart failure, and cardiac arrhythmia (mainly AF) [174,175,176]. Altogether, it represents an association of AF with differential gut microbial flora and their related metabolite pattern. The alteration in gut microbiota may be used as a biomarker for AF disease.

### 3.6. Obesity and Type 2 Diabetes Mellitus (T2DM)

Obesity is a global issue that occurs due to the lower consumption of energy in comparison with intake. The risk of T2DM-type multifactorial disease is increased in the case of obesity. Metabolites have enough competence to be a factor in the host–microbiota relations. Researchers have used a combined metagenomic and metabolomic methodology to explore the associations between the alterations in the gut microbial flora and the onset of metabolic malformations in obesity and diabetes [177]. The most consistent phylum associated with obesity is *Proteobacteria,* and the *Firmicutes*/Bacteroides ratio usually increases in obese individuals [178]. Due to restricted diets like low-calorie, low-fat, or low- carbohydrate, in individuals aiming for weight loss, the *Firmicutes* and *Bacteroidetes* ratio is reduced [178,179]. Duodenal microbiomes, such as *Bifidobacterium* and *Lactobacillus,* were also identified in obese persons, and their observed diversity was different when compared with that of the overweight person, which indicates their significant therapeutic potential. Metagenomics data support the connection between the alteration in gut bacterial composition in obesity and T2DM. In T2DM, butyrate-producing bacteria diminished and enhanced the *Lactobacillus* spp. [180,181,182,183,184]. Another study also suggested that glucose tolerance impairment is associated with T2DM; vancomycin decreases the butyrate-producing bacteria in metabolic syndrome patients and diminishes insulin sensitivity [181,182,183,184,185]. Hosomi et al. (2022) [186] reported that the *Blautia* genus, mainly *B. wexlerae,* is inversely associated with obesity and T2DM. The impacts of these bacteria are interrelated with unique regulatory pathways. It helps in the production of acetylcholine, S-adenosylmethionine, and L-ornithine (amino acid metabolism), the accumulation of amylopectin (carbohydrate metabolism), and production of succinate, lactate, and acetate [186]. Other researchers found many bacterial taxa, including *Acidaminococcales*, *Bacteroides plebeius*, and *Phascolarctobacterium* sp. CAG207 bacterial species significantly differ in obese patients with T2DM and healthy controls [187]. T2DM individuals have lower levels of hydrogen sulfide. Sulfate-reducing bacteria, such as *Desulfovibrio and Desulfomonas*, produce hydrogen sulfide, a product of protein fermentation that affects the basal hepatic glucose levels [188].

Cardiovascular insulin resistance (CVIR) refers to the reduced capability of cardiovascular tissues, specifically vascular smooth muscle cells, endothelial cells, and cardiomyocytes, to respond to insulin. This dysfunction occurs in two ways: “direct and indirect”. Direct dysfunction happens within the heart or vasculature itself; however, indirect dysfunction results from systemic insulin resistance in liver, muscle, and adipose tissues. One molecular paradox is PI3K-Akt that promotes nitric oxide production, vasodilation, anti-inflammatory effects [189,190].

## 4. Gut Microbiota and Therapeutic Potential

Based on numerous studies, vast evidence indicates and supports gut microbiota and its metabolites as a critical factor in the onset of cardiovascular diseases (Table 2).

Thus, gut bacteria might be used as a biomarker and an ideal target for disease prevention, diagnosis, and treatment. A key strategy in maintaining the composition of these bacteria in the gut is dietary intervention. By adopting heart-healthy diets rich in fiber and green vegetables, individuals can take control of their health and influence their gut microbiota. These diets should also include fruits, legumes, and unsaturated fats. Dietary fibers are particularly helpful in producing SCFAs and maintaining the composition of beneficial gut microbes [191]. Antibiotics also play a role in modulating gut microflora to prevent CVDs. In one study, chronic and resistant hypertensive patients were cured after antibiotic treatment [192].

Additionally, after using vancomycin, the myocardial infarct area was reduced, and post-ischemic recovery improved with a reduction in different intestinal microbial groups. Recently, it has been reported that imidazole propionate (ImP), produced by microorganisms, might be explored as an atherosclerosis target [193]. Dietary fiber food like chickpeas and whole-grain oats enhances the bacterial dysbiosis by altering the relative abundance of *Bacteroides* and *Lactobacillus*. Chickpeas help maintain gut bacterial composition and prevent hyperglycemia, whereas whole-grain oats control insulin sensitivity and cholesterol levels [194,195]. Prebiotics are nonmicrobial bodies supplied to stimulate a promising effect on microbial population structure and function. They primarily include dietary fibers, non-digestible molecules, or plant-based supplement food containing oligosaccharides and polysaccharides, which help increase friendly gut microbes. Numerous studies have reported that prebiotics impact host metabolism and have beneficial anti-CVD effects through lipid control, glycemic control, reduced endotoxemia and inflammation, and maintained blood pressure [196,197]. Oligofructose supplementation helps obese patients lose weight and improve glucose tolerance [198,199,200,201,202]. On the other hand, probiotics include beneficial microbes determined to establish the usual gut microbiome equilibrium. One meta-analysis study reported that probiotic treatment may help lower the total cholesterol as well as the low-density lipoprotein (LDL) cholesterol levels, improve blood pressure, and modulate inflammatory cytokines. The most significant probiotic species are *Bifidobacterium* spp. and *Lactobacillus* spp., and at the same time, the most common probiotics are yogurt, kefir, sauerkraut, and kimchi in human diets. As with prebiotics, probiotics also help improve CVDs, providing reassurance and confidence in their use. *Lactobacillus plantarum* helps in the improvement of endothelium-dependent vasodilation [203], *Lactobacillus fermentum* CECT5716 treatment improves endothelial function and controls vascular oxidative stress [204], and yogurt helps in blood glucose and antioxidant level maintenance [205]. *Bifidobacterium longum* BB536 can potentially diminish the total cholesterol and liver lipid deposition levels and adipocyte size [206]. *Akkermansia muciniphila* reduces atherosclerotic lesions [207]. Other probiotics are *Christensenella minuta*, which protects from obesity, and *Lactobacillus reuteri*, which improves insulin secretion [208,209]. Symbiotic is a blend of prebiotics and probiotics designed to promote the growth as well as the survival of beneficial microorganisms. Numerous studies on animals like chickens and mice reported the benefits of symbiotics for improving gut microbes’ composition, immune functions, and modifications in body weight and metabolic syndrome [210,211,212]. Another therapeutic approach is FMT (fecal microbiota transplantation), in which a patient’s intestinal microbes are replaced by beneficial microbes from healthy subjects to treat gastrointestinal diseases [213]. There are two types of transfers: one is autologous transfer, in which the same individual’s feces microbiota is used, and another is allogenic transfer, in which other healthy individual feces microbiota is transferred. FMT is more beneficial in humans with *Clostridium difficile* antibiotic-resistant disease. It is currently tested against cardiometabolic disorders [214]. The limitation of FMT is that it is associated with risks, including the possibility of transferring endotoxins or infectious agents that may cause new complications. Thus, more research is warranted to guarantee the effectiveness and safety of FMT.

## 5. Conclusions and Future Perspectives

Human diets modify human health, and numerous animal and human studies support communalism bacteria. Recent metagenomics and metabolomics approaches, next-generation high-throughput sequencing, and bioinformatic tools help to uncover the different untapped gene pools of gut microbes, metabolites, and their effects on human health that were not reported earlier. Moreover, it is important to characterize the bacterial metabolites that play a role in the host’s physiological modifications to treat cardiometabolic disorders. Numerous microbial pathways are involved directly or indirectly in the stimulation of CVDs. Different host pathways attract microbes and help alter the composition of microbial metabolites that may cause CVDs. More attention is needed to understand how gut microbes convert dietary and endogenous molecules into metabolites and their connections with different tissues and organs of the host. It is also essential to gather some knowledge about the roles of microbial protein interactions, modifications, and post-translational modifications and their impacts and connections with host proteins related to CVDs. Alterations in intestinal microbe structures and functions through dietary interventions, prebiotics, probiotics, symbiotics, antibiotics, and FMT have the potential to alter host metabolism as needed. In the early diagnosis or as therapeutic targets of CVDs, gut microbiota shows their potential. Future research should focus on finding the exact bacteria linked to each type of cardiovascular disease and exploring the substances these bacteria produce that may affect the disease. This could help in the identification of new metabolomic biomarkers of diseases and further our understanding of the role of gut microbiota in CVDs.

## Figures and Tables

**Figure 1 cells-14-01237-f001:**
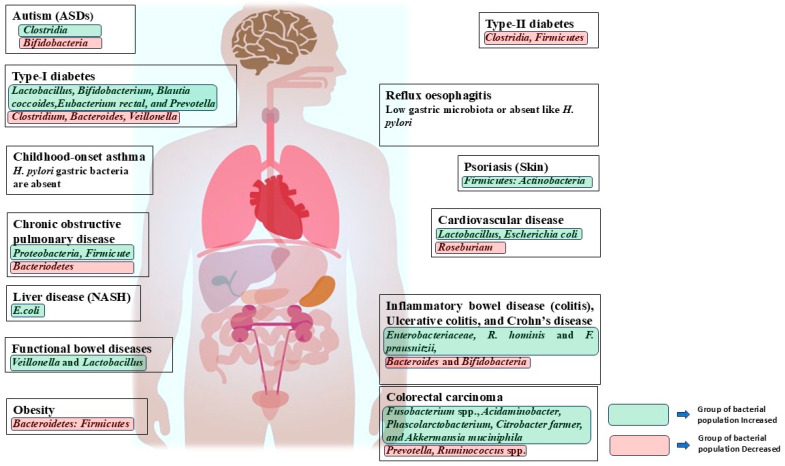
Dysbiosis of gut microbiota and related human diseases.

**Figure 2 cells-14-01237-f002:**
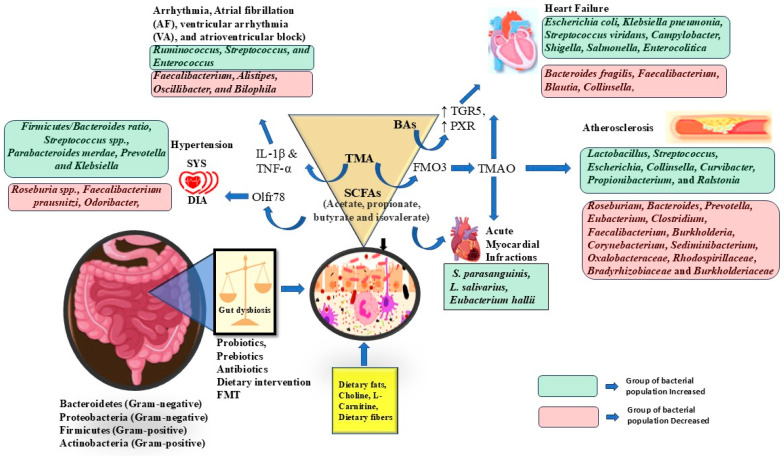
Gut microbiota and metabolite’s role in cardiovascular diseases.

**Table 1 cells-14-01237-t001:** Key gut microbiome-derived metabolites and their functions and roles in CVDs.

Metabolite	Microbiota	Examples	Functions	Mechanisms	Role in CVDs
SCFAs	*Butyricimonas*, *Akkermansia*, *Bacteroides*, *Prevotella*, *Enterolactone Anaerostipes*, *Blautia*, *Coprococcus*, *Eubacterium*, *Faecalibacterium*, *Marvinbryantia*, *Megasphaera*, *Roseburia*, *Ruminococcus*	Acetate, propionate, butyrate	Regulate immune responses and inflammation, maintain gut barrier integrity, influence glucose and lipid metabolism	Act via G-protein-coupled receptors (e.g., GPR41, GPR43) and inhibits histone deacetylases (HDACs)	Excessive or dysregulated SCFA production (due to dysbiosis) are potentially harmful. Their role in heart failure, and hypertension
TMAO	*Collinsella*, *Escherichia*, *Shigella*, *Enterococcus* *Micrococcus*, *Mobiluncus* *Clostridium*, *Staphylococcus*, *Sarcina*, *Campylobacter*, *Pseudomonas*, *Anaerococcus*, *Desulfovibrio*, *Edwardsiella*, *Proteus*, *Providencia*	Trimethylamine (TMA), oxidized in the liver to TMAO	Promotes atherosclerosis, associated with cardiovascular disease (CVD) risk	Alters cholesterol metabolism and enhances platelet aggregation	TMAO increases atherosclerosis and heart failure
Aromatic Amino Acids (AAAs)	*Bacteroides*, *Bifidobacterium*, *Clostridium*, *Lactobacillus*, *Peptostreptococcus*, *Ruminococcus*, *Ruminiclostridium*	Indole, indole-3-acetic acid (IAA), indole-3-propionic acid (IPA), phenylacetylglutamine	Modulate mucosal immunity via aryl hydrocarbon receptor (AhR), influence gut-brain axis and neuroinflammation	Act via G-protein-coupled receptors	Role of indoxyl sulfate in aortic calcification and increased carotid intima media thickness
Bile acids (BAs)	*Butyricimonas*, *Akkermansia*, *Bacteroides*, *Prevotella*, *Enterolactone*	Deoxycholic acid (DCA), lithocholic acid (LCA)	Regulate lipid and glucose metabolism, modulate signaling via FXR and TGR5 receptors	Dysregulation linked to insulin resistance and NAFLD	Role in myocardial fibrosis

**Table 2 cells-14-01237-t002:** CVD-related key gut microbiota and related metabolites.

Diseases	Microbiota	Metabolites
Atherosclerosis	*Enterobacteriaceae*, *Ruminococcus gnavus*, *Eggerthella lenta*, *Roseburia intestinalis*, *Faecalibacterium* cf., *prausnitzii*, *Lactobacillus*, *Streptococcus*, *Clostridium subcluster*, *Bacteroides*	Increase TMAO
Myocardial infarction	*Tissierella soehngenia genera, Lactobacillus plantarum 299v*, *Bacteroides fragilis*	TMAO and SCFAs
Heart failure	*Faecalibacterium prausnitzii*, *Bacteroides fragilis*, *Escherichia coli*, *Klebsiella pneumonia*, *Streptococcus viridans*, *Campylobacter*, *Candida*, *Shigella*, *Salmonella*, *Yersinia enterocolitica*	Increase TMAO
Arrhythmia	*Ruminococcus*, *Streptococcus*, *Enterococcus*, *Faecalibacterium*, *Alistipes*, *Oscillibacter*, *Bilophila*	Role of TMAO in arrythmia
Hypertension	*Firmicutes*, *Bacteroides*, *Prevotella*, *Erwinia*, *Corynebacteriac-eae*, *Anaerostipes*, *Lactobacillus murinus*, *Roseburia intestinalis*	Increase SCFAs

## Data Availability

No new data were created or analyzed in this study.

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
