# Peer review of "Microbial Metabolites and Cardiovascular Dysfunction: A New Era of Diagnostics and Therapy"

_cells, 2025, doi:10.3390/cells14161237_

Round 1
Reviewer 1 Report
Comments and Suggestions for Authors
This review manuscript summarises what is currently known about the links between various elements of the gut biome and a range of common cardiovascular diseases. Given the growing awareness of the potential importance of gut-systemic interactions in health and disease, the review is important and timely. It is likely that the title will raise expectations and generate considerable interest to a wide readership.
However, the topic is complex, and most readers will be hearing of the many organisms mentioned and how they interactions with specific nutritional intake and the entero-systemic absorptions pathways of metabolites for the first time. In its current form, the article will probably not be accessible or have the impact it could for the following reasons:
- Many sections read like catalogues of names of organisms, referenced to individual biochemical interactions, but without apparently any overarching structure. Paragraphs are not internally coherent and would benefit from restructuring.
- Some sections seem contradictory, with initial sentences indicating possible beneficial interactions with certain organisms only to end describing deleterious effects through the same general biome-systemic interactions (eg: SCFA section).
- The article seems too long, and the author has not succeeded in making the topic accessible to the wide clinical readership attracted to learning more about the relevance of the gut biome to their specialty (ie: cardiovascular health and disease). In its current form the review will only be intelligible to researchers already active in the field.
- A shorter, more concise structure, illustrating some of the more validated gut biome-systemic interactions rather than detailing all, would be more informative and maintain reader interest better.
- I would have liked to see the author take the reader from what they are likely to already know and provide them with a ‘briefing’ on what they should be aware of about gut biome-cardiovascular interactions.
- In terms of persuasiveness, it is unclear from the repeated use of words such as 'associated', ‘may be’, ‘closely linked to’ in many sections how credible the effects described actually are (ie: distinguishing evidence from speculation). The review should indicate more clearly the levels of evidence associated with the various reactions described.
- The author does not address the important topic of whether / to what extent the gut biome, once established, can be altered, if at all. That is an important omission for a clinical readership – particularly so, as the article points out how alterations in gut microflora can have deleterious consequences for health.
- Although it seems plausible that differences between individuals in their gut biome may be a factor favouring maintenance of health or contributing to common cardiovascular disorders, the links presented seem unconvincing and disjointed. A plethora of other factors are already known to contribute to the pathophysiology of these complex cardiovascular disorders. The article does not indicate the relative weight of effect of the gut biome effect in relation to more conventional risk factors.
- Disappointingly, the overall impression is that the article is ‘a tough read’, describing apparently random biochemical interactions between gut microorganisms and the nutrients they encounter and concluding with speculative ways in which biproducts might help maintain health or promote disease.
The article makes appropriate use of English throughout.
The work is unnecessarily long and poorly structured with examples of circumlocution and seems to have some internal contradictions.
Author Response
- Many sections read like catalogues of names of organisms, referenced to individual biochemical interactions, but without apparently any overarching structure. Paragraphs are not internally coherent and would benefit from restructuring.
Reply:
We have addressed names of organisms as per the scientific nomenclature and reframed the sentences.
- Some sections seem contradictory, with initial sentences indicating possible beneficial interactions with certain organisms only to end describing deleterious effects through the same general biome-systemic interactions (eg: SCFA section).
Reply:
The review majorly highlights the beneficial aspects of the microbiota of gut in CVDs. However, there are no absolute beneficial bacterial populations in wild. Hence, the negative aspect was included for the overall comparison in gut microbiota. However, as per suggestion, we have reframed the SCFA section. While SCFAs are initially presented as beneficial microbial metabolites, the section concludes by underscoring their potential deleterious effects—particularly in dysbiosis contexts. This inconsistency arises from a lack of nuance in distinguishing host-specific and contextual variables (e.g., microbial composition, immune status, metabolic conditions) that modulate SCFA impact. Therefore, although the duality of SCFA function is biologically plausible, the framing lacks clarity in separating generalized benefits from conditional drawbacks.
- The article seems too long, and the author has not succeeded in making the topic accessible to the wide clinical readership attracted to learning more about the relevance of the gut biome to their specialty (ie: cardiovascular health and disease). In its current form the review will only be intelligible to researchers already active in the field.
Reply:
The review is comprehensive in scope, its length and technical density limit its accessibility for the broader clinical audience interested in gut biome implications for cardiovascular health. The article assumes a high level of prior knowledge, which may alienate readers outside of microbiome research. To make the content more impactful we have rephrased the sentences and reduced the length of the review.
- A shorter, more concise structure, illustrating some of the more validated gut biome-systemic interactions rather than detailing all, would be more informative and maintain reader interest better.
Reply:
The manuscript would benefit from a more concise structure that emphasizes well-established gut biome-systemic interactions over exhaustive enumeration of all potential mechanisms. We have made it shorter and concise and reduced the length of the review.
- I would have liked to see the author take the reader from what they are likely to already know and provide them with a ‘briefing’ on what they should be aware of about gut biome-cardiovascular interactions.
Reply:
The review could be significantly strengthened by guiding readers through a progressive narrative—from foundational concepts they are likely familiar with to emerging insights about gut biome-cardiovascular interactions. By providing a clear ‘briefing’ that bridges conventional cardiovascular understanding with microbiome science.
- In terms of persuasiveness, it is unclear from the repeated use of words such as 'associated', ‘may be’, ‘closely linked to’ in many sections how credible the effects described actually are (ie: distinguishing evidence from speculation). The review should indicate more clearly the levels of evidence associated with the various reactions described.
Reply:
We have removed the words and rephrased the sentences.
- The author does not address the important topic of whether / to what extent the gut biome, once established, can be altered, if at all. That is an important omission for a clinical readership – particularly so, as the article points out how alterations in gut microflora can have deleterious consequences for health.
Reply:
The review primarily focuses on the role of an already-established human gut microbiota in influencing cardiovascular health. It effectively outlines how specific bacterial groups contribute to disease development by modulating chemical cues and cellular pathways associated with CVDs. However, it does not adequately address whether or to what extent the gut microbiota can be altered once established—a crucial omission given the clinical implications of microbiome manipulation. Highlighting the potential for intervention through diet, probiotics, or therapeutics would strengthen its relevance for clinicians.
- Although it seems plausible that differences between individuals in their gut biome may be a factor favouring maintenance of health or contributing to common cardiovascular disorders, the links presented seem unconvincing and disjointed. A plethora of other factors are already known to contribute to the pathophysiology of these complex cardiovascular disorders. The article does not indicate the relative weight of effect of the gut biome effect in relation to more conventional risk factors.
Reply:
This review provides the insight into the role of gut microbiota in regulation of CVDs. We do not encourage it as the major parameter in causing the disease but as an additional factor.
- Disappointingly, the overall impression is that the article is ‘a tough read’, describing apparently random biochemical interactions between gut microorganisms and the nutrients they encounter and concluding with speculative ways in which biproducts might help maintain health or promote disease.
Reply:
We respectfully disagree with the reviewer’s characterization. The biochemical interactions described in the manuscript are not random, but rather reflect the complex and dynamic interplay between gut microorganisms and dietary components. These interactions are influenced by environmental factors, lifestyle, and nutrition, and are directly implicated in the dysregulation of metabolic pathways that contribute to the onset and progression of cardiovascular diseases (CVDs). The review delineates how microbial metabolites affect host physiology by modulating cellular signaling, immune responses, and metabolic regulation. Far from speculative, these associations are supported by emerging evidence, with references provided throughout to substantiate the functional role of microbial byproducts in health and disease.
- Fang X, Zhang Y, Huang X, Miao R, Zhang Y, Tian J. Gut microbiome research: Revealing the pathological mechanisms and treatment strategies of type 2 diabetes. Diabetes Obes Metab. 2025 Aug;27(8):4051-4068. doi: 10.1111/dom.16387. Epub 2025 Apr 15. PMID: 40230225.
- Mastrangelo, A., Robles-Vera, I., Mañanes, D. et al.Imidazole propionate is a driver and therapeutic target in atherosclerosis. Nature (2025). https://doi.org/10.1038/s41586-025-09263-w
Comments on the Quality of English Language
The article makes appropriate use of English throughout.
Reply:
Improved
The work is unnecessarily long and poorly structured with examples of circumlocution and seems to have some internal contradictions.
Reply:
We appreciate the reviewer’s feedback regarding structure and clarity. The breadth of content was intended to reflect the multifaceted nature of gut microbiota-cardiovascular interactions, a field that continues to evolve.
Reviewer 2 Report
Comments and Suggestions for Authors
This review is updated and potentially interesting. However, it is relatively too long and includes some typos and redundant sentences. In addition, some abbreviations are repeated.
For example, "impacting lower blood pressure" should be "impacting with blood pressure regulation". Later on in the same paragraph "protect from the risk of hypertensive cardiovascular damage and aortic atherosclerotic lesions" should be protect from the risk of vascular abnormalities and aortic atherosclerotic lesions"
Please check the manuscript for English, abbreviations, repetitions and contents.
Please also clearly state the type of the manuscript at the end of the introduction and in the conclusive sections. It appears to be a narrative review, thus selection of references should be potentially biased. This should be clearly acknowledged by the author.
Comments on the Quality of English LanguageThis review is updated and potentially interesting. However, it is relatively too long and includes some typos and redundant sentences. In addition, some abbreviations are repeated.
For example, "impacting lower blood pressure" should be "impacting with blood pressure regulation". Later on in the same paragraph "protect from the risk of hypertensive cardiovascular damage and aortic atherosclerotic lesions" should be protect from the risk of vascular abnormalities and aortic atherosclerotic lesions"
Please check the manuscript for English, abbreviations, repetitions and contents.
Please also clearly state the type of the manuscript at the end of the introduction and in the conclusive sections. It appears to be a narrative review, thus selection of references should be potentially biased. This should be clearly acknowledged by the author.
Author Response
- This review is updated and potentially interesting. However, it is relatively too long and includes some typos and redundant sentences. In addition, some abbreviations are repeated. For example, "impacting lower blood pressure" should be "impacting with blood pressure regulation". Later on in the same paragraph "protect from the risk of hypertensive cardiovascular damage and aortic atherosclerotic lesions" should be protect from the risk of vascular abnormalities and aortic atherosclerotic lesions"
Reply:
Correction done
- Please check the manuscript for English, abbreviations, repetitions and contents.
Reply:
Correction done
- Please also clearly state the type of the manuscript at the end of the introduction and in the conclusive sections. It appears to be a narrative review, thus selection of references should be potentially biased. This should be clearly acknowledged by the author.
Reply:
Correction done
Round 2
Reviewer 1 Report
Comments and Suggestions for Authors
This manuscript provides a comprehensive overview of current understanding of the links between an individuals gut microbiome and the development / prevention of a range of common cardiovascular conditions. The topic is of considerable and growing interest to clinicians. However, it is a complex one, mainly because the organisms involved are unknown to most clinicians and the multitude of ways in which gut organisms and their metabolic products interact with nutrients and the body is unfamiliar to all but those working in this area of research.
This redraft of the manuscript has improved the submission significantly. Sections are now better explained, the logic flow of paragraphs is improved and all sub-topics are particularly well referenced. The authors have succeeded in making the complex topic of the relationship of gut-biome to health and disease much more accessible to the average readership. I particularly like the way the figures have been modified so that they are now largely self explanatory.
Occasional sentences which would benefit from editing for word usage and punctuation but the sense even of these is still clear.